# The Comparison of Different Methods of Texture Analysis for Their Efficacy for Land Use Classification in Satellite Imagery

**Przemysław Kupidura**

Faculty of Geodesy and Cartography, Warsaw University of Technology, 00-661 Warsaw, Poland; przemyslaw.kupidura@pw.edu.pl

**Abstract:** The paper presents a comparison of the efficacy of several texture analysis methods as tools for improving land use/cover classification in satellite imagery. The tested methods were: gray level co-occurrence matrix (GLCM) features, Laplace filters and granulometric analysis, based on mathematical morphology. The performed tests included an assessment of the classification accuracy performed based on spectro-textural datasets: spectral images with the addition of images generated using different texture analysis methods. The class nomenclature was based on spectral and textural differences and included the following classes: water, low vegetation, bare soil, urban, and two (coniferous and deciduous) forest classes. The classification accuracy was assessed using the overall accuracy and kappa index of agreement, based on the reference data generated using visual interpretation of the images. The analysis was performed using very high-resolution imagery (Pleiades, WorldView-2) and high-resolution imagery (Sentinel-2). The results show the efficacy of selected GLCM features and granulometric analysis as tools for providing textural data, which could be used in the process of land use/cover classification. It is also clear that texture analysis is generally a more important and effective component of classification for images of higher resolution. In addition, for classification using GLCM results, the Random Forest variable importance analysis was performed.

**Keywords:** satellite imagery; classification; texture analysis; GLCM; mathematical morphology; granulometric analysis; Laplace filter

## 1. Introduction

Texture is one of the most important spatial features of an image. Compared to other important spatial features, such as shape and size, it is relatively simple to use because it does not require prior image segmentation. At the same time, it is a distinctive feature of selected land use/cover classes, compared to other classes exhibiting significant spectral similarities. For example, urban and bare soil areas share similar spectral characteristics, as do forests and areas of low vegetation. As the research shows, the use of textural information in classification, apart from spectral data, can significantly increase the accuracy of classification [1–12]. The best results can be obtained by using a combination of spectral and textural data [7,8,12].

Texture has no unambiguous definition, which is why in the practice of digital image processing there are many different methods of texture analysis defined ad hoc. Some of these methods include gray level co-occurrence matrix (GLCM) [1,2,13], fractal analysis [3], discrete wavelet transformation [14], Laplace filters [15–17], Markov random fields [18,19] or granulometric analysis [20,21]. There are also studies showing the high potential of artificial neural networks, including convolutional ones, for spectral-spatial approaches to classification [5,6].

The following paper presents a comparison of the effectiveness in providing textural information of GLCM, Laplacian and granulometric analyses. The first two methods are relatively well researched, also in terms of the effectiveness of textural analysis. However, granulometric analysis is a lesser-known method. Although previous studies [4,7,8] show its significant potential, there are no studies comparing it with other methods of textural analysis. The main motivation of this paper is therefore to present such a comparative analysis.

Previous studies [7] show that spatial resolution is important when identifying textural signatures which indicate a specific classification of coverage or land use. It was shown that the significance of the texture decreases with the spatial resolution of the image and that it is not important in the case of images with a pixel of approximately 30 m. Therefore, this study used images with different resolutions: very high (GSD (ground sample distance) 2 m: Pleiades and WorldView-2) and high: (GSD 10 m: Sentinel-2).

## 2. Brief Presentation of Tested Methods of Textural Analysis

Three methods of textural analysis were tested: GLCM, Laplace filters and granulometric analysis. They are presented below.

### 2.1. Gray Level Co-Occurrence Matrix (GLCM)

This method, first presented by Julesz [13], is based on creating a matrix describing the frequency of the appearance of individual pairs of values in a specific image fragment (gray level co-occurrence matrix). Then certain features describing certain textura; aspects are calculated. A significant part of these features was developed by Haralick et al. [1,2], thus the indicators are often referred to as Haralick features. Various authors propose the use of various Haralick features [10–12]. The effectiveness of this popular method has been demonstrated in a significant number of publications [22,23]. In this paper, a set of eight different GLCM indicators is applied (formulas according to [24]):

$$Energy = \sum_{i,j} g(i,j)^2 \tag{1}$$

$$Entropy = \sum_{i,j} g(i,j) log_2 g(i,j), \; or \; 0 \; if \; g(i,j) = 0, \tag{2}$$

$$Correlation = \sum_{i,j} \frac{(i-\mu)(j-\mu)g(i,j)}{\sigma^2}, \tag{3}$$

$$Inverse\ Difference\ Moment = \sum_{i,j} \frac{1}{1+(i-j)^2} g(i,j), \tag{4}$$

$$Inertia = \sum_{i,j} (i,j)^2 g(i,j), \tag{5}$$

$$Cluster\ Shade = \sum_{i,j} ((i-\mu)+(j-\mu))^3 g(i,j), \tag{6}$$

$$Cluster\ Prominence = \sum_{i,j} ((i-\mu)+(j-\mu))^4 g(i,j), \tag{7}$$

$$Correlation = \sum_{i,j} \frac{(i-\mu)(j-\mu)g(i,j)}{\sigma^2}, \tag{8}$$

where $(i,j)$ is the matrix cell index, $g(i,j)$ is the frequency value of the pair having index $(i,j)$, $\mu = \sum_{i,j} i * g(i,j) = \sum_{i,j} j * g(i,j)$ (due to matrix symmetry) and means weighted pixel average, $\sigma = \sum_{i,j} (i-\mu)^2 * g(i,j) = \sum_{i,j} (j-\mu)^2 * g(i,j)$ (due to matrix symmetry) and means weighted pixel variance, and $\mu_t$ and $\sigma_t$ are the mean and standard deviation of the row (or column, due to symmetry) sums.

*2.2. Laplace Filters*

Laplacian filters are derivative filters used to find areas of rapid change in coincident imagery. They have been presented in [25,26]. Laplace filters can be expressed using a convolution [26] e.g., using a mask as presented in the Figure 1.

| -1 | -1 | -1 |
|----|----|----|
| -1 | 8  | -1 |
| -1 | -1 | -1 |

**Figure 1.** Exemplary mask of Laplace filter, used in the presented research.

They are often used to detect edges of objects in an image. They can also be used to detect the parts of an image with high texture, characterized by a high spatial frequency [27–29]. It can give good results compared to other similar methods, such as the Sobel filter, but also in comparison with Haralick's features [29].

*2.3. Granulometric Analysis*

The third method, granulometric analysis, is not well-known, although its effectiveness has also been demonstrated in previous publications [4,7,8,30]. It resembles a morphological profile [31,32], although at the same time it differs significantly from it in some respects [7,8].

Granulometric analysis is based on the sequence of morphological opening and closing operations and the measurement of the differences between successive images. This permits the quantification of particles of different sizes [7]. The method was first presented by Haas, Matheron and Serra [20]. However, methods of local analysis were introduced later [21], allowing the assignment of texture values to individual pixels. Its accuracy, regarding use in the classification of satellite imagery, has been demonstrated in previous studies [7,8]. Granulometric analysis can be based on classical (simple) morphological operations of opening and closing, as well as on operations with a multiple structuring element (MSE) [7]. As shown by the studies, both these versions of granulometric analysis show slightly different properties. Depending on the image and distinguished land use/cover classes, differing results may be obtained [7].

As this method is relatively unknown, the two basic advantages of this texture analysis method are briefly described below.

The first is multiscality; due to the possibility of successive application of increasing size of morphological opening and closing operations, the obtained information indicates the presence of texture grains of various sizes.

The second advantage is resistance to the so-called edge effect [7,33]. The edge effect means that the edges of objects, even those with a low texture, get high values as a result of texture analysis. This applies to most textural analysis methods because they refer to the spatial frequency analysis of the selected image area as a texture determinant. Imagery edges have a high spatial frequency, and thus are exhibited with high texture. Granulometric analysis is not based on this principle, as it analyzes the number and value of removed image elements. Because of this, edges are not display as areas of high texture.

## 3. Material and Methods

The study consisted of processing selected satellite multispectral imagery of high and very high resolution using the tested methods of texture analysis, then combining the results of individual methods with original spectral images and finally classifying such datasets and assessing their accuracy.

### 3.1. Source Spectral Data

In this study, images showing the areas the South-East of Warsaw (Poland) were used. This is an area characterized by diversified land cover. There are, among others, agricultural land, coniferous and deciduous forests, water reservoirs and various forms of buildings. This study used images with different resolutions: GSD 2 m (Pleiades and WorldView-2) and 10 m (Sentinel-2). These were subsets of satellite scenes. Details of the test images are shown in Table 1. The images are shown in Figure 2.

**Table 1.** Test images used in the study.

| Test Image | Satellite Platform | GSD | Spectral Bands | Date of Acquisition |
|---|---|---|---|---|
| 1 | Pleiades | 2 m | blue, green, red, near infrared | 22.05.2012 |
| 2 | WorldView-2 | 2 m | coastal, blue, green, yellow, red, red edge, 2x near infrared | 04.08.2011 |
| 3 | Sentinel-2 | 10 m | blue, green, red, near infrared | 20.04.2018 |

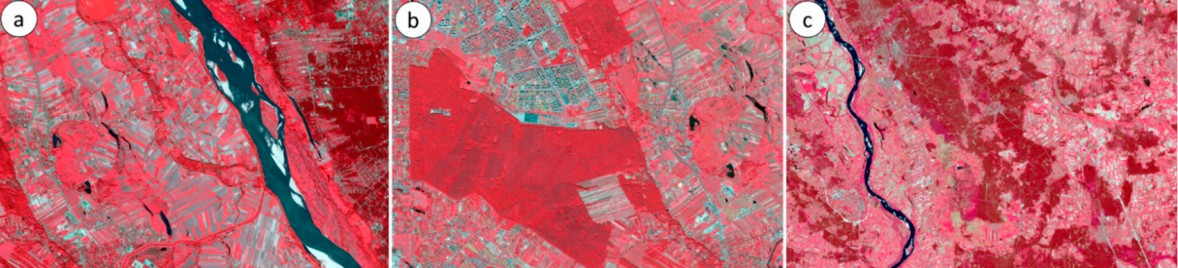

**Figure 2.** Test images: (**a**) 1—Pleiades, (**b**) 2—WorldView-2, (**c**) 3—Sentinel-2.

### 3.2. Textural Data

The research involved four methods of texture analysis: GLCM, Laplace filter and granulometric analysis (based on simple operations and operations with multiple structuring elements, MSE). All texture images were obtained based on the processing of the image of the first principal component. The choice of the first principal component was based on previous studies showing that the use of this image as the source date for texture analysis gives the best results when it comes to separability of selected land use classes compared to other images such as second principal components or selected spectral channels [34]. Each of the multispectral images has been subjected to principal component analysis. Then, the images of the first components, which by definition represent the largest variance within the analyzed multispectral data, were subjected to textural processing using the tested methods. As a result, three basic data sets were prepared, presented in Table 2.

**Table 2.** The set of textural images.

| Texture Analysis Method | Images | Number of Images |
|---|---|---|
| GLCM | Results of GLCM (gray level co-occurrence matrix) features presented in Section 2.1: Energy, Entropy, Correlation, Inverse Difference Moment, Inertia, Cluster Shade, Cluster Prominence, Haralick's Correlation. | 8 |
| Laplace filters | Results of Laplacian of size 1, 2 and 3 ($3 \times 3$, $5 \times 5$, $7 \times 7$) | 3 |
| Granulometric analysis | Three granulometric maps based on simple morphological opening and three granulometric maps based on simple morphological closing (two for each in the case of test image 3) | 6 (4 for test image 3) |
| Granulometric analysis basing on operations with multiple structuring element (MSE) | Three granulometric maps based on morphological MSE opening and three granulometric maps based on morphological MSE closing (two for each in the case of test image 3) | 6 (4 for test image 3) |

In the case of granulometric analysis, a different number of subsequent granulometric maps (being the result of openings and closures of successively larger sizes of the structuring element) were used, depending on the spatial resolution of the image. In the case of higher resolution photos (GSD: 2 m), i.e., test image 1 and test image 2, these were three consecutive granulometric maps for opening and closing, while in the case of test image 3 (GSD: 10 m), these were two consecutive granulometric maps for opening and closing.

Analyses with selected methods (GLCM and both versions of granulometric analysis) were carried out in several variants, depending on the size of the analyzed neighborhood of individual pixels. These were areas with 5, 7, 10 and 13 pixels.

### 3.3. Methodology

As part of the research, a series of classifications were performed on each of the test images. These were classifications made on different sets of data consisting of spectral image-only data and on sets of spectro-textural data, enriched with the results of textural analysis, obtained on the basis of selected methods. The tested variants are listed and explained in Table 3.

**Table 3.** Classification variants.

| Name of the Variant | Spectral Data | Textural Data |
| --- | --- | --- |
| *spectral* | Yes | None |
| *spectral + Laplacian* | Yes | Laplace filters |
| *spectral + GLCM5* | Yes | 8 GLCM features, neighborhood: size 5 |
| *spectral + GLCM7* | Yes | 8 GLCM features, neighborhood: size 7 |
| *spectral + GLCM10* | Yes | 8 GLCM features, neighborhood: size 10 |
| *spectral + GLCM13* | Yes | 8 GLCM features, neighborhood: size 13 |
| *spectral + gran5* | Yes | 6 (or 4) simple granulometric maps, neighborhood: size 5 |
| *spectral + gran7* | Yes | 6 (or 4) simple granulometric maps, neighborhood: size 7 |
| *spectral + gran10* | Yes | 6 (or 4) simple granulometric maps, neighborhood: size 10 |
| *spectral + gran13* | Yes | 6 (or 4) simple granulometric maps, neighborhood: size 13 |
| *spectral + MSEgran5* | Yes | 6 (or 4) MSE granulometric maps, neighborhood: size 5 |
| *spectral + MSEgran7* | Yes | 6 (or 4) MSE granulometric maps, neighborhood: size 7 |
| *spectral + MSEgran 10* | Yes | 6 (or 4) MSE granulometric maps, neighborhood: size 10 |
| *spectral + MSEgran13* | Yes | 6 (or 4) MSE granulometric maps, neighborhood: size 13 |

The classification was performed using the random forest [35] method based on training fields developed on the basis of a multispectral image. The classifier contained 500 trees, the number of features was equal to the square root of all features and the impurity function was based on the Gini coefficient.

In all variants of the classification, exactly the same training fields were used. For each test image a relatively large number of training fields (from 65 to 74, then aggregated to the final number of classes) were prepared to ensure the highest possible classification accuracy, so that the differences obtained for individual variants depended only on the type of input data. The six following classes were distinguished during this process:

1.  Water
2.  Bare soil
3.  Low vegetation
4.  Coniferous forest
5.  Deciduous forest
6.  Built-up area

To perform the textural analysis using the selected methods, the image of the first principal component, calculated on the basis of a set of multispectral data, was used. Accuracy assessment was

performed by comparing the results of the classification with the reference image created on the basis of the test sites. The test sites were developed as a result of the visual interpretation of the image. They were to meet the requirements ensuring proper control of the classification: equal distribution over the entire classified area and proportional representation of all classes [36]. The total number of test pixels was large in order to ensure high reliability of the accuracy check (980,869 pixels for Test Image 1, 489,573 pixels for Test Image 2 and 250,952 pixels for Test Image 3; other statistics concerning individual classes may be found in corresponding matrices).

The error matrix was compiled for the result of each classification, and errors of omission (OE) and commission (CE) [37] as well as overall accuracy (OA) and the kappa index of agreement (KIA) [38] were calculated. The scheme of the methodology is shown in Figure 3.

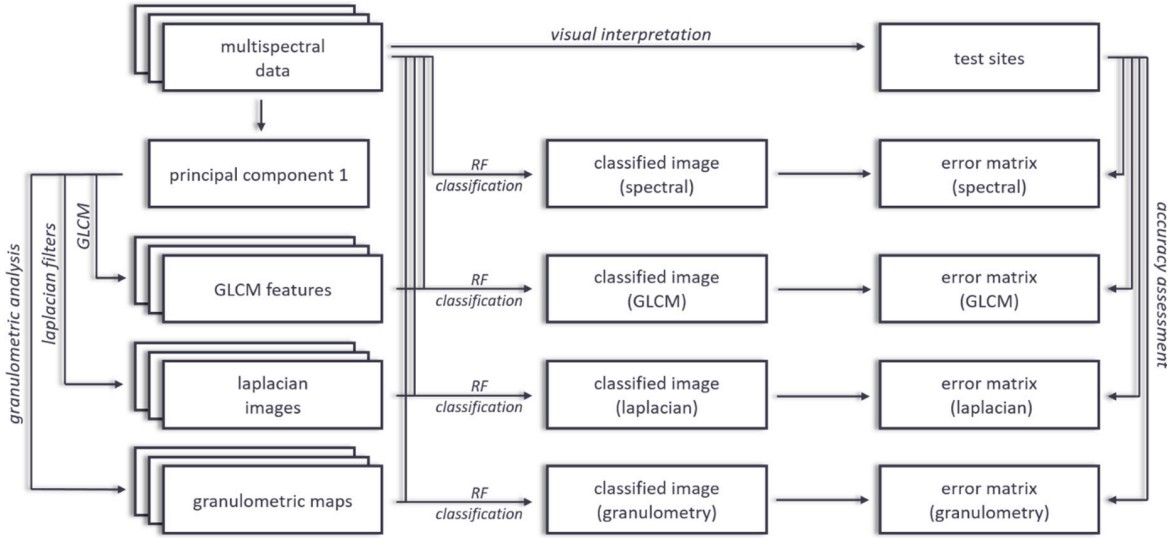

**Figure 3.** The methodology scheme.

In addition, for the classification of the GLCM results, the random forest variables importance analysis was performed. Variable importance is calculated based on out-of-bag accuracy and signifies the importance of the respective variable; a high value means a high importance of the variable for the entire random forest model and vice versa [35,39]. The GLCM data set is the only set of utilized data (in the section on texture images) with images of a qualitatively different nature; different images present different features, which in turn are referring to other aspects of the image's texture. Hence, this analysis was performed, permitting a significant assessment of particular Haralick features used for satellite image classification.

## 4. Results

Individual analyses were carried out on three test images. This section contains a summary and analysis of the results obtained.

### 4.1. Results of Classification

#### 4.1.1. Test image 1—Pleiades (2 m)

The results of the analysis are summarized in Table 4.

The Pleiades multispectral image is characterized by a relatively high spatial resolution (GSD: 2 m). As can be seen in Table 4, the accuracy of the classification based only on spectral data is low (OA: 0.78, KIA: 0.71). In all scenarios in which the results of textural analysis were additionally used, the accuracy of the classification was significantly higher. The best results were obtained for the classification using granulometric maps: *spectral + gran10*, i.e., obtained as a result of an analysis

using simple morphological operations inside a radius of 10 pixels (OA: 0.98, KIA: 0.97). However, it should be noted that for all such operations, regardless of the radius of the analyzed neighborhood, the results were similarly high—the lowest for *spectral + gran5* (OA: 0.96, KIA: 0.94), but still very high— higher than the GLCM or Laplacian results. The results obtained for the spectral + GLCM classification are also relatively good (OA: 0.89–0.92, KIA: 0.86–0.90), but clearly worse than using granulometric analysis.

**Table 4.** Summary of the results for test image 1—Pleiades (2 m).

| Scenario | Overall Accuracy (OA) | Kappa Index of Agreement (KIA) |
|---|---|---|
| *spectral* | 0.78 | 0.71 |
| *spectral + Laplacian* | 0.83 | 0.77 |
| *spectral + GLCM5* | 0.90 | 0.87 |
| *spectral + GLCM7* | 0.92 | 0.90 |
| *spectral + GLCM10* | 0.92 | 0.89 |
| *spectral + GLCM13* | 0.89 | 0.86 |
| *spectral + gran5* | 0.96 | 0.94 |
| *spectral + gran7* | 0.97 | 0.96 |
| *spectral + gran10* | 0.98 | 0.97 |
| *spectral + gran13* | 0.96 | 0.95 |
| *spectral + MSEgran5* | 0.89 | 0.86 |
| *spectral + MSEgran7* | 0.93 | 0.91 |
| *spectral + MSEgran 10* | 0.96 | 0.94 |
| *spectral + MSEgran13* | 0.96 | 0.95 |

Tables 5–8 present the error matrices of the sample classification scenarios. Spectral classification is shown in Table 3. The selected classification images obtained for individual scenarios are presented in Figure 4.

**Table 5.** Error matrix for spectral classification of test image 1, Pleiades.

| | | Reference Image | | | | | | | |
|---|---|---|---|---|---|---|---|---|---|
| | | 1. water | 2. soil | 3. low veg | 4. con. forest | 5. dec. forest | 6. built-up | Σ | CE |
| classification | 1. water | 91,346 | 1 | 1 | 43 | 2 | 1031 | 92,424 | **0.01** |
| | 2. soil | 0 | 248,368 | 796 | 0 | 0 | 2810 | 251,974 | **0.01** |
| | 3. low veg | 0 | 61 | 208,706 | 2 | 4101 | 307 | 213,177 | **0.02** |
| | 4. con. forest | 134 | 4 | 2 | 103,989 | 11,846 | 345 | 116,320 | **0.11** |
| | 5. dec. forest | 0 | 0 | 148,536 | 1119 | 92,983 | 483 | 243,121 | **0.62** |
| | 6. built-up | 0 | 42,047 | 55 | 56 | 1 | 21,694 | 63,853 | **0.66** |
| | Σ | 91,480 | 290,481 | 358,096 | 105,209 | 108,933 | 26,670 | **980,869** | |
| | OE | **0.00** | **0.14** | **0.42** | **0.01** | **0.15** | **0.19** | OA | **0.782** |
| | | | | | | | | KIA | **0.709** |

**Table 6.** Error matrix for classification *spectral + gran10* of test image 1—Pleiades.

| | | Reference Image | | | | | | | |
|---|---|---|---|---|---|---|---|---|---|
| | | 1. water | 2. soil | 3. low veg | 4. con. forest | 5. dec. forest | 6. built-up | Σ | CE |
| classification | 1. water | 91,353 | 151 | 2 | 47 | 0 | 111 | 91,664 | **0.00** |
| | 2. soil | 0 | 285,996 | 233 | 0 | 0 | 1524 | 287,753 | **0.01** |
| | 3. low veg | 3 | 3669 | 351,592 | 10 | 1020 | 535 | 356,829 | **0.01** |
| | 4. con. forest | 124 | 120 | 135 | 104,781 | 6026 | 915 | 112,101 | **0.07** |
| | 5. dec. forest | 0 | 0 | 5811 | 352 | 101,879 | 768 | 108,810 | **0.06** |
| | 6. built-up | 0 | 545 | 323 | 19 | 8 | 22,817 | 23,712 | **0.04** |
| | Σ | 91,480 | 290,481 | 358,096 | 105,209 | 108,933 | 26,670 | **980,869** | |
| | OE | **0.00** | **0.02** | **0.02** | **0.00** | **0.06** | **0.14** | OA | **0.977** |
| | | | | | | | | KIA | **0.969** |

**Table 7.** Error matrix for classification *spectral + GLCM7* of test image 1, Pleiades.

| | | Reference Image | | | | | | | |
|---|---|---|---|---|---|---|---|---|---|
| | | 1. water | 2. soil | 3. low veg | 4. con. forest | 5. dec. forest | 6. built-up | Σ | CE |
| classification | 1. water | 91,417 | 118 | 1820 | 52 | 10 | 391 | 93,808 | **0.03** |
| | 2. soil | 0 | 273,962 | 4741 | 0 | 1 | 2437 | 281,141 | **0.03** |
| | 3. low veg | 0 | 6499 | 307,073 | 1 | 5098 | 158 | 318,829 | **0.04** |
| | 4. con. forest | 5 | 158 | 39 | 104,910 | 476 | 211 | 105,799 | **0.01** |
| | 5. dec. forest | 54 | 7 | 44231 | 222 | 103,303 | 325 | 148,142 | **0.30** |
| | 6. built-up | 4 | 9737 | 192 | 24 | 45 | 23,148 | 33,150 | **0.30** |
| | Σ | 91,480 | 290,481 | 358,096 | 105,209 | 108,933 | 26,670 | **980,869** | |
| | OE | **0.00** | **0.06** | **0.14** | **0.00** | **0.05** | **0.13** | OA | **0.921** |
| | | | | | | | | KIA | **0.897** |

**Table 8.** Error matrix for classification *spectral + Laplacian* of test image 1, Pleiades.

| | | Reference Image | | | | | | | |
|---|---|---|---|---|---|---|---|---|---|
| | | 1. water | 2. soil | 3. low veg | 4. con. forest | 5. dec. forest | 6. built-up | Σ | CE |
| classification | 1. water | 91,025 | 1 | 1 | 65 | 3 | 741 | 91,836 | **0.01** |
| | 2. soil | 0 | 249,230 | 553 | 0 | 0 | 2894 | 252,677 | **0.01** |
| | 3. low veg | 0 | 101 | 249,377 | 2 | 3768 | 358 | 253,606 | **0.02** |
| | 4. con. forest | 452 | 3 | 0 | 104,689 | 8816 | 332 | 114,292 | **0.08** |
| | 5. dec. forest | 3 | 0 | 108,090 | 416 | 96,345 | 445 | 205,299 | **0,53** |
| | 6. built-up | 0 | 41,146 | 75 | 37 | 1 | 21,900 | 63,159 | **0.65** |
| | Σ | 91,480 | 290,481 | 358,096 | 105,209 | 108,933 | 26,670 | **980,869** | |
| | OE | **0.00** | **0.14** | **0.30** | **0.00** | **0.12** | **0.18** | OA | **0.828** |
| | | | | | | | | KIA | **0.770** |

Spectral classification has moderate accuracy (OA: 0.78, KIA: 0.71; Table 5; Figure 4a). As expected, large classification errors can be noticed in classes where a high texture is an important distinction. A large commission error (CE) is noticeable for Class 6: Built-up area (0.66), which is largely due to the allocation of bare soil pixels (Class 2) to this class. The obvious reason for this is the spectral similarity between these two classes. A similar situation can be observed in the case of a pair of classes: deciduous forest (Class 5) and low vegetation (Class 3). In the case of these two classes, there is also at least partial spectral similarity, especially in the case of illuminated parts of tree crowns. This results in a large CE in Class 5: deciduous forest (0.62) and, at the same time, a large OE in Class 3: low vegetation (0.42). When analyzing the classification error matrix in the spectral scenario, it can be noticed that for the relatively low accuracy of this classification, the responsible classes are the two cases discussed above: built-up areas versus bare soil and deciduous forest versus low vegetation. Therefore, it should be expected that classifications based on data sets that also contain the results of the textural analysis should improve accuracy in this area.

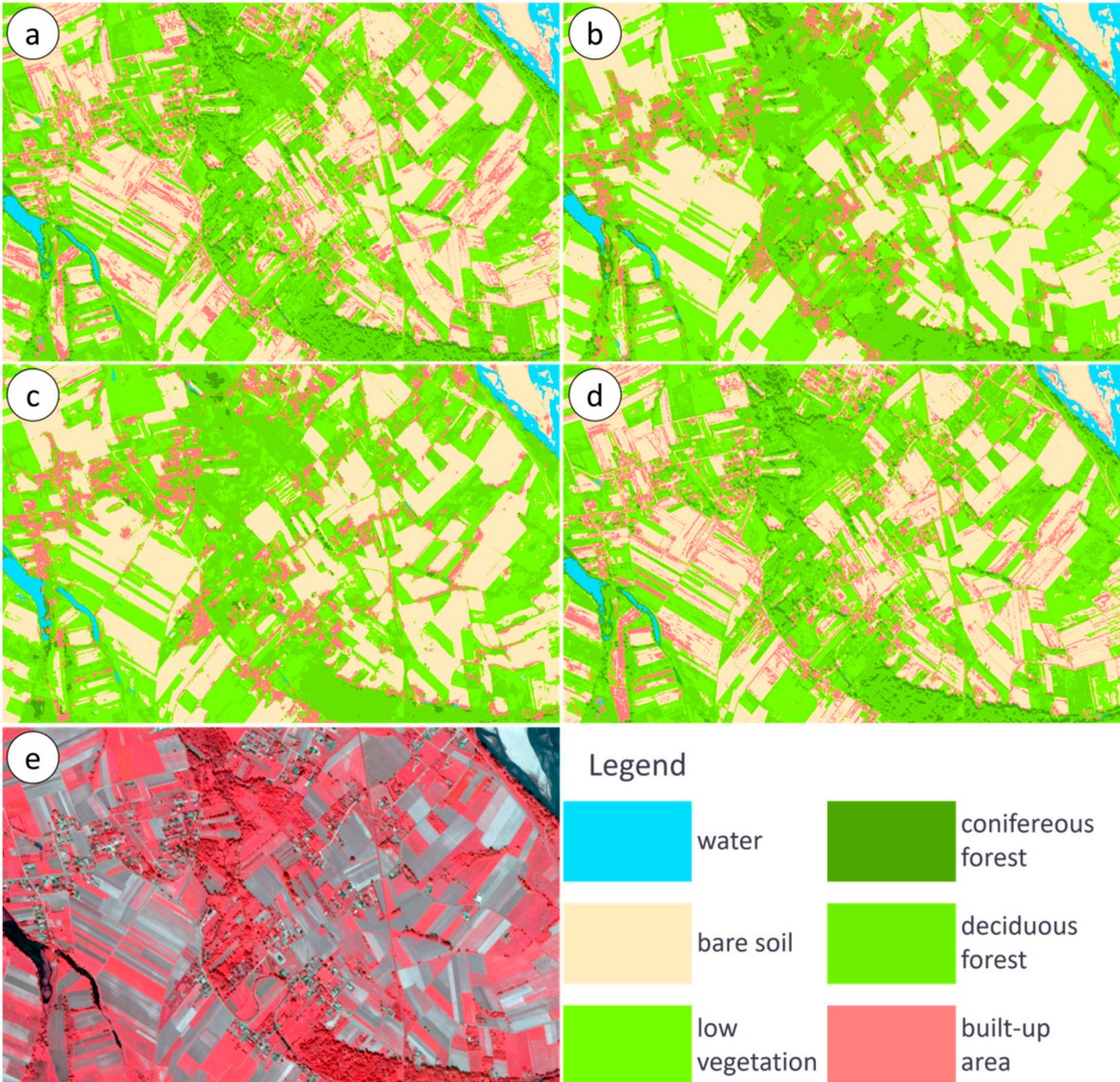

**Figure 4.** Subsets of images of selected classification variants (test image 1, Pleiades): (**a**) *spectral*, (**b**) *spectral + gran10*, (**c**) *spectral + GLCM7*, (**d**) *spectral+ Laplacian*, (**e**) original satellite image.

Table 6 (also Figure 4b) presents the results obtained for the classification using, in addition to spectral data, the results of the granulometric analysis, *spectral + gran10*. This is the classification with the best result of those analyzed for test image 1, Pleiades (OA: 0.98, KIA: 0.97). As expected, this classification significantly improved the separation of the two class pairs (2–6 and 3–5) as compared to the spectral classification, where a large decrease in accuracy was observed. The biggest errors were noted for Class 6, built-up area (OE: 0.14, CE: 0.04). As in the previously analyzed case, they are mainly caused by the incorrect distinction between Class 6 and 2 (bare soil). However, they are much smaller than in the case of spectral classification (OE: 0.19, CE: 0.66). The distinction between classes 2 and 5 also improved considerably due to the granulometric processing; in all other classes, OE and CE values do not exceed 0.07.

The classification based on the spectral data and GLCM (Table 7; Figure 4c) results is also significantly better than spectral classifications, however the obtained accuracy is lower than in the case of the granulometric analysis. For example, errors for class 6 (built-up area), caused mainly by similarity to Class 2 (bare soil) were OE: 0.13 and CE: 0.30, so less than in the spectral classification, but more than in the case of classification *spectral + gran10*. Although classes 2 and 5 continue to show

misclassifications, there is an improvement in relation to the spectral classification. However, results are still less accurate than those resulting from the *spectral + gran10* classification.

The classification using the results of the Laplace transformations gave the worst results among the spectro-textural classification variants (Table 8; Figure 4d). Interestingly, despite the fact that the general classification results have improved in relation to the spectral classification, the results for individual classes are even worse than in the case of the latter. An example may be Class 6, built-up area (OE: 0.18, CE: 0.65). The reason for such a large CE is the assignment to Class 6, bare soil areas (in fact Class 2). These types of misclassification also occurred within other classifications, but not on such a scale. The analysis of the classification images shows clearly that the reason for such a large error is, first and foremost, the erroneously classified pixels of the exposed soil located on the borders of agricultural plots (Figure 4). Here, the edge effect mentioned above is visible.

4.1.2. Test Image 2: WorldView (2 m)

The results of the accuracy assessment for classifications of test image 2: WorldView-2 are presented in Table 9.

**Table 9.** Summary of the results for test image 2—WorldView-2 (2 m).

| Scenario | Overall Accuracy (OA) | Kappa Index of Agreement (KIA) |
|---|---|---|
| *spectral* | 0.94 | 0.92 |
| *spectral + Laplacian* | 0.95 | 0.93 |
| *spectral + GLCM5* | 0.89 | 0.86 |
| *spectral + GLCM7* | 0.88 | 0.85 |
| *spectral + GLCM10* | 0.86 | 0.82 |
| *spectral + GLCM13* | 0.87 | 0.83 |
| *spectral + gran5* | 0.96 | 0.95 |
| *spectral + gran7* | 0.96 | 0.95 |
| *spectral + gran10* | 0.96 | 0.95 |
| *spectral + gran13* | 0.96 | 0.94 |
| *spectral + MSEgran5* | 0.95 | 0.94 |
| *spectral + MSEgran7* | 0.96 | 0.95 |
| *spectral + MSEgran 10* | 0.97 | 0.96 |
| *spectral + MSEgran13* | 0.96 | 0.96 |

Unlike in the previous case, the spectral image classification is characterized by relatively high accuracy (OA: 0.94, KIA: 0.92). Also, unlike before, not all types of textural data improved the accuracy of the classification; the use of data obtained on the basis of the GLCM analysis caused a deterioration of the results. The best result for the GLCM operation was obtained for the *spectral + GLCM5* variant (OA: 0.89, KIA 0.86). The remaining operations allowed greater accuracy; this applies to both the *spectral + Laplacian* variant (OA: 0.95, KIA: 0.93), and both versions of the granulometric operations (simple and MSE). In the latter case, the best results were obtained for the *spectral + MSEgran10* variant (OA: 0.97, KIA: 0.96), however, for all variants based on granulometric data, very similar high results were obtained (e.g., *spectral + gran10*: OA: 0, 96, KIA: 0.95). The differences between the individual variants are therefore insignificant.

More detailed information is provided by the analysis of the error matrix of selected classification variants. These matrices are presented in Tables 10–13. The subsets of selected classification images obtained for individual scenarios are presented in Figure 5.

**Table 10.** Error matrix for spectral classification of test image 2: WorldView-2.

| | | Reference Image | | | | | | | |
|---|---|---|---|---|---|---|---|---|---|
| | | 1. water | 2. soil | 3. low veg | 4. con. forest | 5. dec. forest | 6. built-up | Σ | CE |
| classification | 1. water | 6849 | 1 | 0 | 20 | 3 | 367 | 7240 | **0.05** |
| | 2. soil | 208 | 47,391 | 153 | 373 | 86 | 11194 | 59,405 | **0.20** |
| | 3. low veg | 0 | 27 | 119,853 | 1089 | 781 | 78 | 121,828 | **0.02** |
| | 4. con. forest | 0 | 0 | 902 | 87,820 | 9391 | 65 | 98,178 | **0.11** |
| | 5. dec. forest | 0 | 0 | 1720 | 3406 | 127,290 | 2 | 132,418 | **0.04** |
| | 6. built-up | 113 | 767 | 122 | 7 | 1 | 69,494 | 70,504 | **0.01** |
| | Σ | 7170 | 48,186 | 122,750 | 92,715 | 137,552 | 81,200 | **489,573** | |
| | OE | **0.04** | **0.02** | **0.02** | **0.05** | **0.07** | **0.14** | OA | **0.937** |
| | | | | | | | | KIA | **0.920** |

**Table 11.** Error matrix for classification *spectral + gran10* of test image 2: WorldView-2.

| | | Reference Image | | | | | | | |
|---|---|---|---|---|---|---|---|---|---|
| | | 1. water | 2. soil | 3. low veg | 4. con. forest | 5. dec. forest | 6. built-up | Σ | CE |
| classification | 1. water | 6770 | 8 | 0 | 0 | 0 | 679 | 7457 | **0.09** |
| | 2. soil | 369 | 48,000 | 295 | 5 | 48 | 6111 | 54,828 | **0.12** |
| | 3. low veg | 0 | 51 | 122,051 | 3 | 1048 | 47 | 123,200 | **0.01** |
| | 4. con. forest | 0 | 1 | 87 | 91,932 | 8790 | 276 | 101,086 | **0.09** |
| | 5. dec. forest | 0 | 0 | 317 | 748 | 127,662 | 4 | 128,731 | **0.01** |
| | 6. built-up | 31 | 126 | 0 | 27 | 4 | 74,083 | 74,271 | **0.00** |
| | Σ | 7170 | 48,186 | 122,750 | 92,715 | 137,552 | 81,200 | **489,573** | |
| | OE | **0.06** | **0.00** | **0.01** | **0.01** | **0.07** | **0.09** | OA | **0.961** |
| | | | | | | | | KIA | **0.951** |

**Table 12.** Error matrix for classification *spectral + GLCM7* of test image 2: WorldView-2.

| | | Reference Image | | | | | | | |
|---|---|---|---|---|---|---|---|---|---|
| | | 1. water | 2. soil | 3. low veg | 4. con. forest | 5. dec. forest | 6. built-up | Σ | CE |
| classification | 1. water | 6050 | 0 | 1007 | 0 | 0 | 126 | 7183 | **0.16** |
| | 2. soil | 993 | 47,563 | 2291 | 0 | 28 | 12,939 | 63,814 | **0.25** |
| | 3. low veg | 0 | 30 | 118,113 | 5 | 1453 | 53 | 119,654 | **0.01** |
| | 4. con. forest | 0 | 3 | 10 | 90,369 | 35,470 | 189 | 126,041 | **0.28** |
| | 5. dec. forest | 0 | 6 | 1324 | 2336 | 100,598 | 34 | 104,298 | **0.04** |
| | 6. built-up | 127 | 584 | 5 | 5 | 3 | 67,859 | 68,583 | **0.01** |
| | Σ | 7170 | 48,186 | 122,750 | 92,715 | 137,552 | 81,200 | **489,573** | |
| | OE | **0.16** | **0.01** | **0.04** | **0.03** | **0.27** | **0.16** | OA | **0.879** |
| | | | | | | | | KIA | **0.847** |

**Table 13.** Error matrix for classification *spectral + Laplacian* of test image 2: WorldView-2.

| | | Reference Image | | | | | | | |
|---|---|---|---|---|---|---|---|---|---|
| | | 1. water | 2. soil | 3. low veg | 4. con. forest | 5. dec. forest | 6. built-up | Σ | CE |
| classification | 1. water | 6920 | 0 | 0 | 27 | 4 | 364 | 7315 | **0.05** |
| | 2. soil | 195 | 47,588 | 255 | 331 | 80 | 10,643 | 59,092 | **0.19** |
| | 3. low veg | 0 | 73 | 121,952 | 498 | 812 | 91 | 123,426 | **0.01** |
| | 4. con. forest | 1 | 0 | 110 | 88,345 | 8955 | 57 | 97,468 | **0.09** |
| | 5. dec. forest | 0 | 0 | 432 | 3514 | 127,700 | 2 | 131,648 | **0.03** |
| | 6. built-up | 54 | 525 | 1 | 0 | 1 | 70,043 | 70,624 | **0.01** |
| | Σ | 7170 | 48,186 | 122,750 | 92,715 | 137,552 | 81,200 | **489,573** | |
| | OE | **0.03** | **0.01** | **0.01** | **0.05** | **0.07** | **0.14** | OA | **0.945** |
| | | | | | | | | KIA | **0.930** |

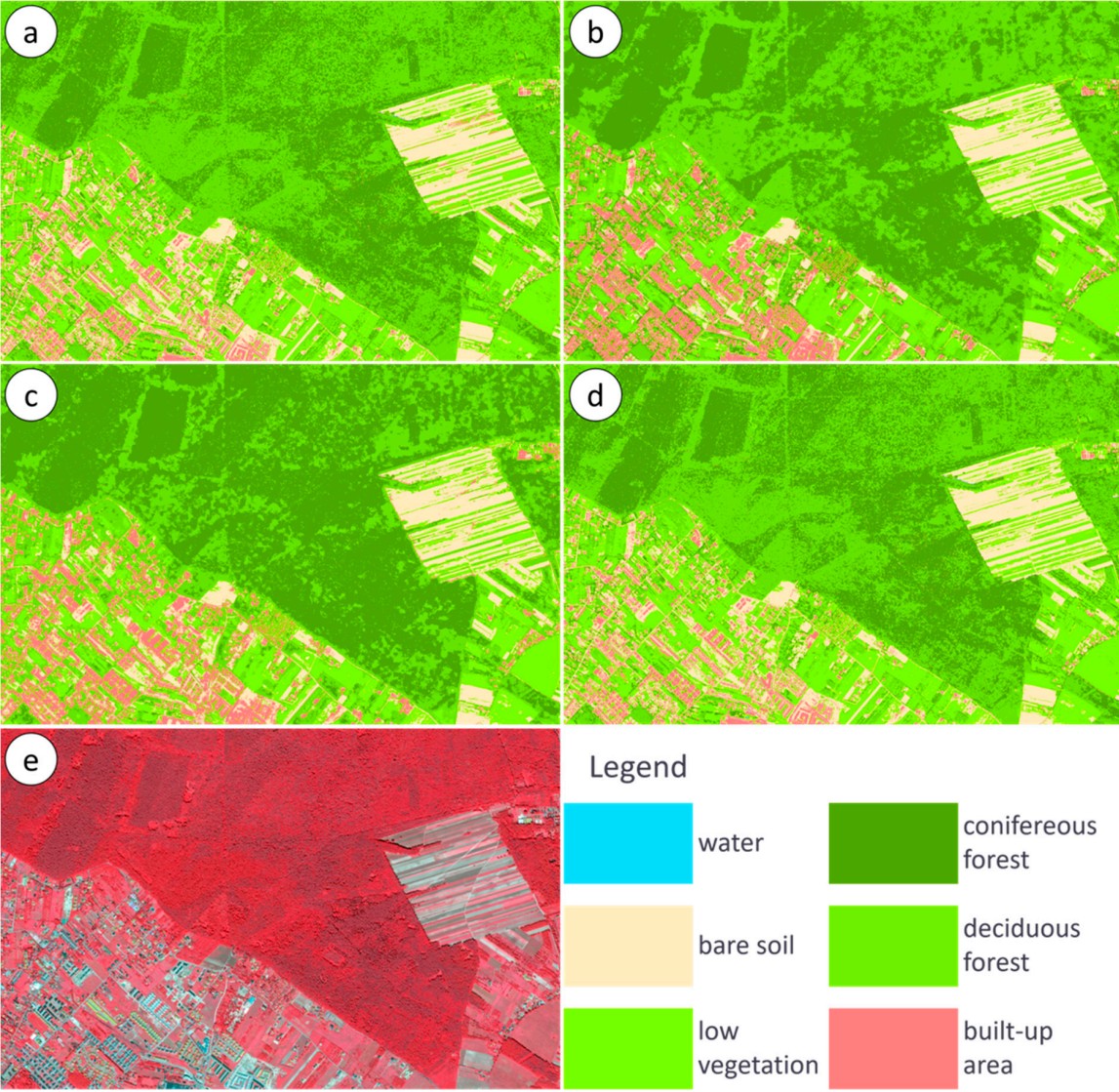

**Figure 5.** Subsets of images of selected classification variants (test image 2: WorldView-2): (**a**) *spectral*, (**b**) *spectral + gran10*, (**c**) *spectral + GLCM7*, (**d**) *spectral+ Laplacian*, (**e**) original satellite image.

The spectral classification of test image 2 (Table 10; Figure 5a) is characterized by a high degree of accuracy (OA: 0.95, KIA: 0.92), much higher than in the case of test image 1. Because the pixel size of both analyzed images is the same, the reasons for these differences should be sought primarily at the time of acquisition of the image. It was taken in August, the period that falls, at least partly, after harvesting, which means that the total area of completely bare soil is relatively small: in some cases, there is the presence of post-harvest residues that change the soil's spectral characteristics sufficiently to significantly improve the distinction of bare soil from built-up areas. This is reflected in relatively small errors for Class 2 (bare soil, OE: 0.02, CE: 0.20) and 6 (built-up area, OE: 0.14, CE: 0.01). The accuracy of forest classification is also higher than in the previous case. This time, larger errors were obtained for Class 4 (coniferous forest, OE: 0.05, CE: 0.11), mainly due to the erroneous allocation of deciduous forest areas.

The accuracy of the classification obtained thanks to the additional use of granulometric data (Table 11; Figure 5b) is the largest of all general types of variants, as in the case of test image 1. The improvement in comparison to the spectral classification is smaller (as in the case of test image 1), but it results from the already high accuracy of the spectral variant. Thus, it is possible to observe a reduction in errors in all classes (the exception is the slight decrease in the accuracy of Class 1, water).

The classification based on the results of the GLCM analysis (Table 12; Figure 5c) gave surprisingly poor results, much worse than the spectral classification. Analysis of the error matrix shows that the main reasons for this are due to Class 4 (coniferous forest, OE: 0.03, CE: 0.28) and 5 (deciduous forest, OE: 0.27, CE: 0.04). These errors are mainly due to the allocation of a significant part of the pixels representing deciduous forest to Class 4, coniferous forest. There is also a slight decrease (in relation to the spectral classification) of the accuracy of Class 2 (bare soil) and 6 (built-up area). The accuracy of Class 1 (water) is also noticeable (OE: 0.16, CE: 0.16), as shown in the analysis of classification images, mainly at the edges of water areas. The case of this classification seems to suggest that if the distinction between individual classes based on spectral data is fairly accurate (see the results of spectral classification), additional data may hinder the classification and deteriorate its results.

The use of the Laplace filtering results slightly increased the accuracy of classification (Table 13; Figure 5d). This is due to a slight improvement in the designation of all classes.

### 4.1.3. Test Image 3—Sentinel-2 (10 m)

The results of the accuracy assessment for classifications of test image 3, Sentinel-2 are presented in Table 14.

**Table 14.** Summary of the results for test image 3: Sentinel-2 (10 m).

| Classification Variant | Overal Accuracy (OA) | Kappa Index of Agreement (KIA) |
|:---:|:---:|:---:|
| *spectral* | 0.93 | 0.90 |
| *spectral + Laplacian* | 0.92 | 0.90 |
| *spectral + GLCM5* | 0.95 | 0.93 |
| *spectral + GLCM7* | 0.95 | 0.93 |
| *spectral + GLCM10* | 0.94 | 0.92 |
| *spectral + GLCM13* | 0.94 | 0.92 |
| *spectral + gran5* | 0.97 | 0.96 |
| *spectral + gran7* | 0.97 | 0.96 |
| *spectral + gran10* | 0.98 | 0.97 |
| *spectral + gran13* | 0.97 | 0.96 |
| *spectral + MSEgran5* | 0.98 | 0.97 |
| *spectral + MSEgran7* | 0.97 | 0.96 |
| *spectral + MSEgran 10* | 0.97 | 0.96 |
| *spectral + MSEgran13* | 0.97 | 0.96 |

The accuracy of the spectral classification is relatively high (OA: 0.93, KIA 0.90). Apart from the *spectral + Laplacian* variant, all variants using textural analysis results are nonetheless more accurate. The best overall result was obtained for the *spectral + gran10* variant (OA: 0.98, KIA: 0.97), among the GLCM variants it was the *spectral + GLCM7* variant (OA: 0.95, 0.93). Therefore, these are the same variants that gave the best results in the case of test image 1. More detailed information is provided by the analysis of the error matrix of the selected classification variants. These matrices are presented in Tables 15–18. The subsets of selected classification images obtained for individual scenarios are presented in Figure 6.

**Table 15.** Error matrix for spectral classification of test image 3: Sentinel-2.

| | | Reference Image | | | | | | | |
|---|---|---|---|---|---|---|---|---|---|
| | | 1. water | 2. soil | 3. low veg | 4. con. forest | 5. dec. forest | 6. built-up | Σ | CE |
| classification | 1. water | 13,671 | 0 | 0 | 0 | 1 | 21 | 13,693 | **0.00** |
| | 2. soil | 0 | 70,477 | 265 | 0 | 2 | 2156 | 72,900 | **0.03** |
| | 3. low veg | 4 | 28 | 61,144 | 171 | 717 | 922 | 62,986 | **0.03** |
| | 4. con. forest | 25 | 0 | 0 | 59,225 | 337 | 28 | 59,615 | **0.01** |
| | 5. dec. forest | 0 | 0 | 1067 | 1197 | 16,568 | 31 | 18,863 | **0.12** |
| | 6. built-up | 1 | 11,489 | 182 | 2 | 2 | 11,219 | 22,895 | **0.51** |
| | Σ | 13,701 | 81,994 | 62,658 | 60,595 | 17,627 | 14,377 | **250,952** | |
| | OE | **0.00** | **0.14** | **0.02** | **0.02** | **0.06** | **0.22** | OA | **0.926** |
| | | | | | | | | KIA | **0.902** |

**Table 16.** Error matrix for classification *spectral + gran10* of test image 3: Sentinel-2.

| | | Reference Image | | | | | | | |
|---|---|---|---|---|---|---|---|---|---|
| | | 1. water | 2. soil | 3. low veg | 4. con. forest | 5. dec. forest | 6. built-up | Σ | CE |
| classification | 1. water | 13,620 | 0 | 104 | 0 | 0 | 24 | 13,748 | **0.01** |
| | 2. soil | 0 | 81,178 | 243 | 0 | 3 | 725 | 82,149 | **0.01** |
| | 3. low veg | 4 | 263 | 61,449 | 166 | 712 | 292 | 62,886 | **0.02** |
| | 4. con. forest | 73 | 0 | 220 | 58,598 | 122 | 13 | 59,026 | **0.01** |
| | 5. dec. forest | 4 | 0 | 642 | 1831 | 16,789 | 281 | 19,547 | **0.14** |
| | 6. built-up | 0 | 553 | 0 | 0 | 1 | 13,042 | 13,596 | **0.04** |
| | Σ | 13,701 | 81,994 | 62,658 | 60,595 | 17,627 | 14,377 | **250,952** | |
| | OE | **0.01** | **0.01** | **0.02** | **0.03** | **0.05** | **0.09** | OA | **0.975** |
| | | | | | | | | KIA | **0.967** |

**Table 17.** Error matrix for classification *spectral + GLCM7* of test image 3: Sentinel-2.

| | | Reference Image | | | | | | | |
|---|---|---|---|---|---|---|---|---|---|
| | | 1. water | 2. soil | 3. low veg | 4. con. forest | 5. dec. forest | 6. built-up | Σ | CE |
| classification | 1. water | 13,596 | 0 | 0 | 120 | 41 | 137 | 13,894 | **0.02** |
| | 2. soil | 0 | 77,496 | 295 | 0 | 4 | 1442 | 79,237 | **0.02** |
| | 3. low veg | 0 | 935 | 61,720 | 365 | 2305 | 547 | 65,872 | **0.06** |
| | 4. con. forest | 90 | 3 | 115 | 59,816 | 2290 | 2 | 62,316 | **0.04** |
| | 5. dec. forest | 15 | 4 | 520 | 294 | 12,986 | 274 | 14,093 | **0.08** |
| | 6. built-up | 0 | 3556 | 8 | 0 | 1 | 11,975 | 15,540 | **0.23** |
| | Σ | 13,701 | 81,994 | 62,658 | 60,595 | 17,627 | 14,377 | **250,952** | |
| | OE | **0.01** | **0.05** | **0.01** | **0.01** | **0.26** | **0.17** | OA | **0.947** |
| | | | | | | | | KIA | **0.930** |

**Table 18.** Error matrix for classification *spectral + Laplace* of test image 3: Sentinel-2.

| | | Reference Image | | | | | | | |
|---|---|---|---|---|---|---|---|---|---|
| | | 1. water | 2. soil | 3. low veg | 4. con. forest | 5. dec. forest | 6. built-up | Σ | CE |
| classification | 1. water | 13,662 | 0 | 0 | 7 | 52 | 10 | 13,731 | **0.01** |
| | 2. soil | 0 | 68,946 | 316 | 0 | 2 | 1859 | 71,123 | **0.03** |
| | 3. low veg | 6 | 44 | 61,273 | 141 | 652 | 1025 | 63,141 | **0.03** |
| | 4. con. forest | 32 | 0 | 0 | 59,687 | 150 | 20 | 59,889 | **0.00** |
| | 5. dec. forest | 1 | 0 | 1031 | 758 | 16,771 | 58 | 18,619 | **0.10** |
| | 6. built-up | 0 | 13,004 | 38 | 2 | 0 | 11,405 | 24,449 | **0.53** |
| | Σ | 13,701 | 81,994 | 62,658 | 60,595 | 17,627 | 14,377 | **250,952** | |
| | OE | **0.00** | **0.16** | **0.02** | **0.01** | **0.05** | **0.21** | OA | **0.923** |
| | | | | | | | | KIA | **0.900** |

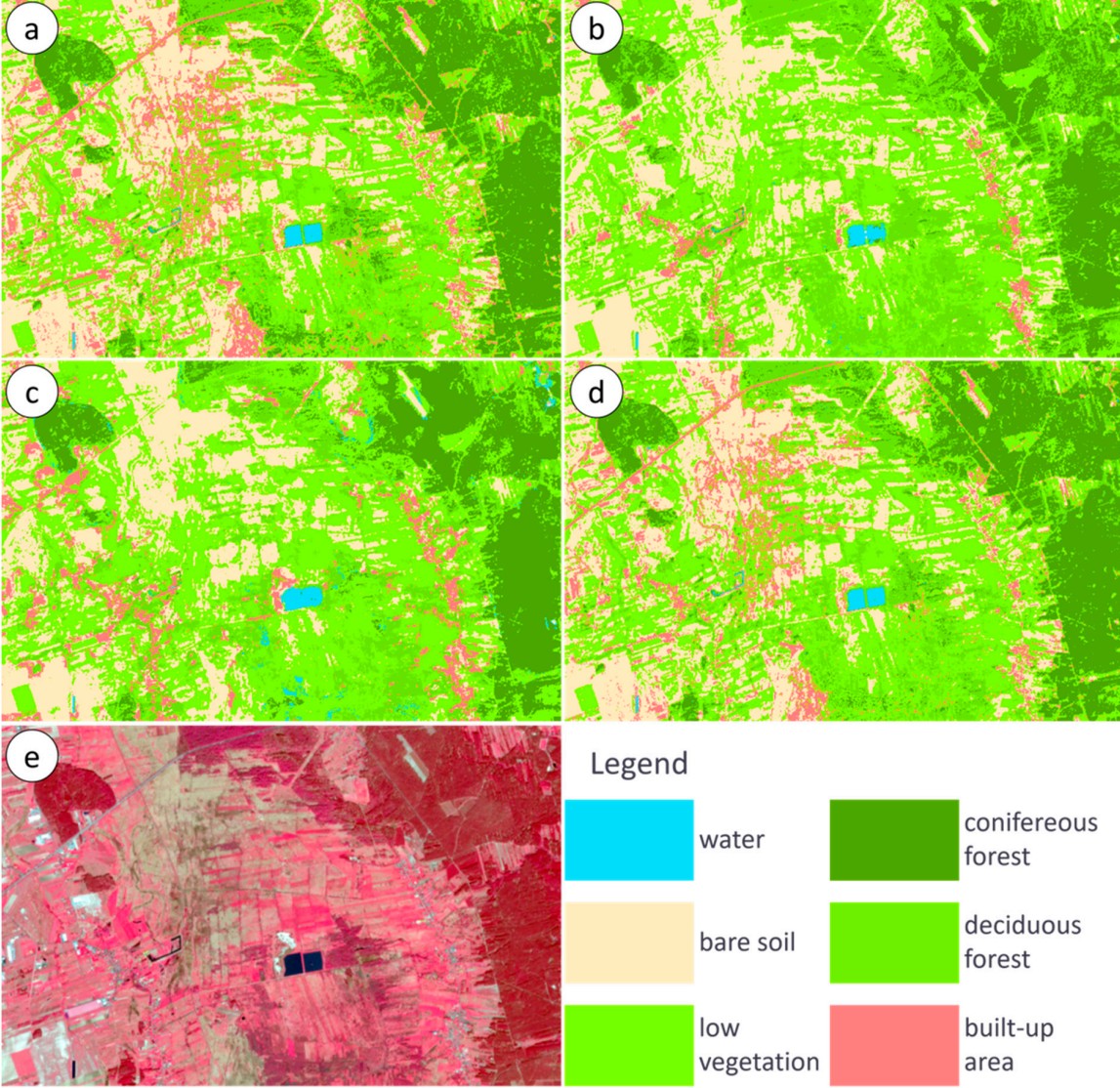

**Figure 6.** Subsets of images of selected classification variants (test image 3: Sentinel-2): (**a**) *spectral*, (**b**) *spectral + gran10*, (**c**) *spectral + GLCM7*, (**d**) *spectral+ Laplacian*, (**e**) original satellite image.

As the analysis of the error matrix for spectral classification (Table 15; Figure 6a) shows, the biggest errors are generated by a problem with the differentiation between Class 6 (built-up area, OE: 0.22: CE: 0.51) and Class 2 (bare soil, OE: 0.14, CE: 0.03). This indicates a high spectral similarity between both land use classes. Other classes are determined with high accuracy, and except for the example of Class 5 (deciduous forest, OE: 0.06, CE: 0.12), which is mainly due to problems with distinguishing pixels of this class from coniferous forest and low vegetation, errors do not exceed the value of 0.03.

The *spectral + gran10* variant is the best of the analyzed ones (Table 16; Figure 6b). The improvement in accuracy is mainly due to an improvement in the classification of Class 6 (built-up area, OE: 0.09, CE: 0.04). Improvements have also been observed in the other classes, e.g., in Class 5, deciduous forest (OE: 0.06, CE: 0.12).

The *spectral + GLCM7* variant (Table 17; Figure 6c) also improved the results relative to the spectral classification, although to a lesser extent than the *spectral + gran10* classification. The increase in accuracy results mainly from the improvement in the accuracy of Class 6, built-up area (OE: 0.17, CE: 0.23), again, to a lesser extent than in the *spectral + gran10* classification. However, a decrease in the accuracy of Class 5 (deciduous forest, OE: 0.26, CE: 0.08), can be noted. A similar effect for this class was noted during the analysis of the classification of test image 2, also based on GLCM data.

Classification variant *spectral + Laplace* is the only one characterized by lower accuracy than the spectral classification (Table 18; Figure 6d). The differences, however, are small. Similarly, the impact of particular classes on the overall accuracy of the classification appear similar. Differences between analogous errors in both variants do not exceed 0.02. It is therefore difficult to say that the use of this data in the classification of the Sentinel-2 photo could have any significant impact on the accuracy of the classification.

*4.2. Analysis of Random Forest Variables Importance for the Dataset Consisting of GLCM Features*

This section presents an analysis of the significance of individual images that make up the *spectral + GLCM* data sets for the random forest classification. This analysis was performed on these data sets only, because they are the only ones among the analyzed variants (in the section on texture images) with images of a qualitatively different nature: different images present different features, which in turn refer to other aspects of the image texture. The analysis allowed the assessment of the significance of particular Haralick features used for satellite image classification.

The diagrams showing the importance of different variables; spectral (marked with gray) and GLCM (marked with black) are shown in Figures 7–9.

In the case of the classification of test image 1: Pleiades (2 m), Haralick's correlation has the greatest significance among the GLCM features. The importance of this layer is similar to the importance of spectral images. The remaining GLCM features are of low importance, at a similar level.

Additionally, in the case of the classification of test image 2: WorldView-2 (2 m), Haralick's correlation shows the greatest importance among all texture images, and except for in one case, had the greatest importance in general. The relatively small significance of individual spectral ranges may in this case result from their greater number; this meaning is distributed among individual images.

Once again, for the classification of test image 3: Sentinel-2, Haralick's Correlation is the most important variable among the GLCM features. However, this significance is relatively less than the other two test images. The explanation of this situation may be a smaller spatial resolution of the image.

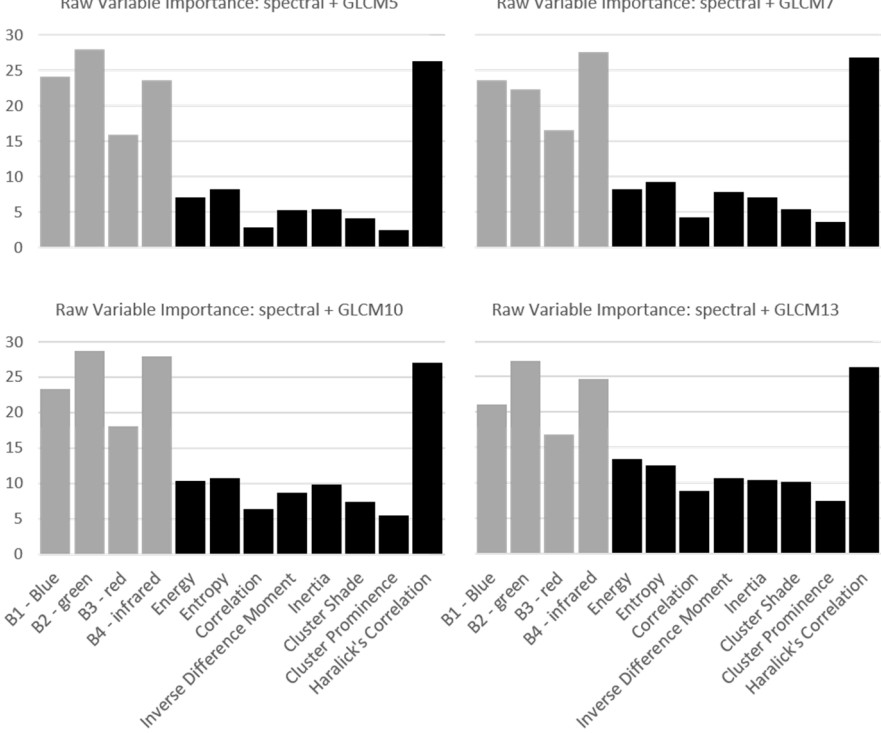

**Figure 7.** Raw variable importance for spectral and GLCM variants, test image 1: Pleiades.

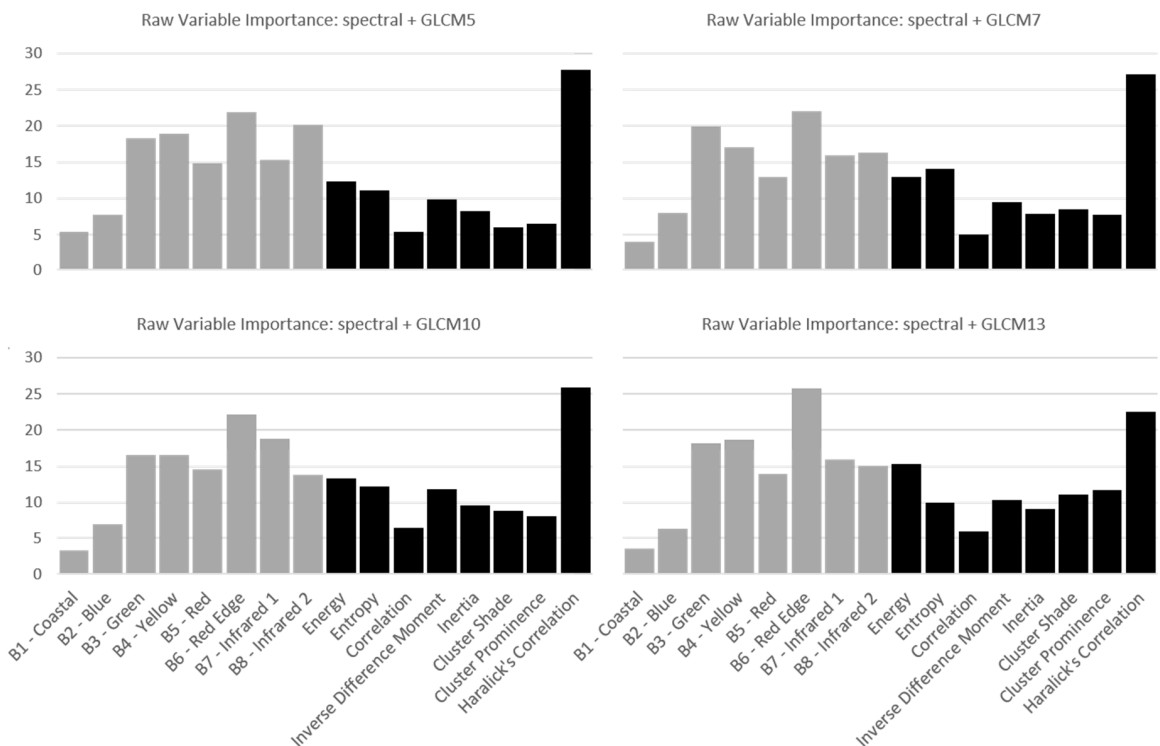

**Figure 8.** Raw variable importance for spectral and GLCM variants, test image 2: WorldView-2.

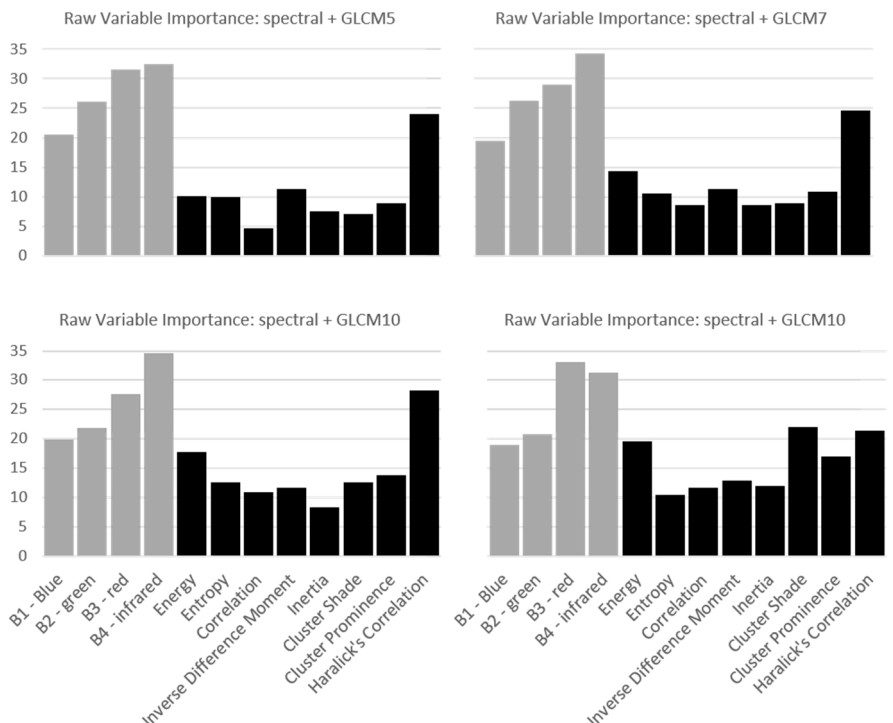

**Figure 9.** Raw variable importance for spectral and GLCM variants, test image 3: Sentinel-2.

## 5. Discussion

The obtained results showed a high efficiency of spectro-textural classification based on the results of granulometric analysis. In all cases, the *spectral + gran* classification variants showed the best accuracy among all analyzed variants. Importantly, the individual *spectral + gran* variants, differing

in the radius of analysis, gave quite similar results. This indicates the high stability of this method. Although it can be stated that the best results were generally obtained with the *spectral + gran10* variant, all variants based on granulometric analysis were significantly better than the other variants of classification—spectral and spectro-textural—based on other texture analysis methods.

This may be at least partly due to the issue of the edge effect, not regarding granulometric processing, but regarding the other two methods tested; GLCM and Laplace filters. This effect is investigated in Figure 10a–c; the edge of the square on the right gets relatively large values in the image as a result of GLCM analysis, similar to the value for a fragment of a high texture image (left side). On the image obtained as a result of granulometric analysis, this effect is not visible. A similar set was prepared for a subset of the actual satellite image of Pleiades (Figure 10d–f). In the GLCM image, the edges of the objects (plots) get high values, falsely indicating a high texture, while on the granulometric map this effect does not occur. This effect can be very important in image classification [7] because it makes parts of objects (on the edges) with a low texture look like objects with a high texture, thus reducing their separability, especially in cases of high spectral similarity.

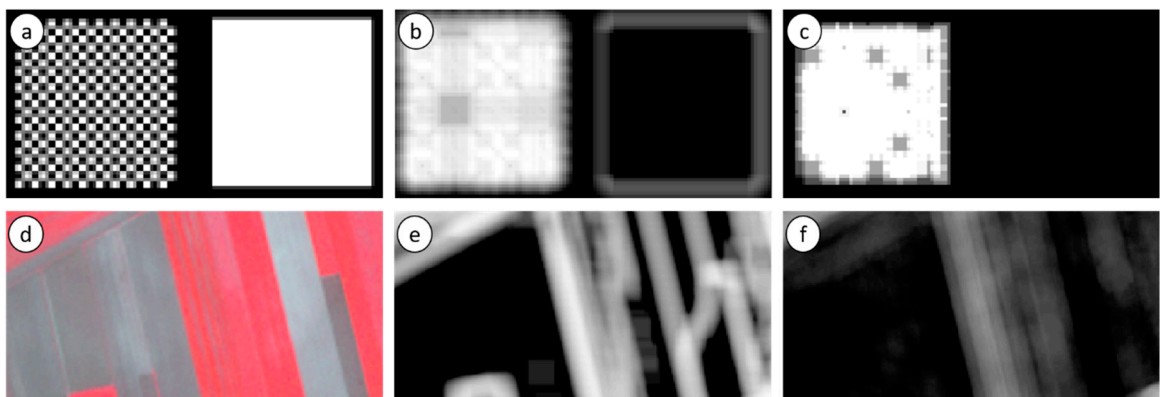

**Figure 10.** Edge effect in simulated imagery: (**a**) original image, (**b**) GLCM entropy, (**c**) granulometric map; and in actual Pleiades image: (**d**) original image, (**e**) GLCM Entropy, (**f**) granulometric map.

We can observe this effect in the example of Test Image 2, where the use of GLCM or Laplace imagery did not improve the distinction between bare soil and built-up areas; it could even (as in the case of GLCM) worsen it. It is worth recalling that due to the date when Test Image 2 was taken, the separability of these classes based on spectral data was relatively large.

The results obtained for Test Image 3 confirm the results of previous studies that with a decrease in spatial resolution, the importance of textural analysis for distinguishing land cover classes decreases. Although, also in this case, the application of the results of the textural analysis increased the accuracy of the classification (with the exception of Laplace's operations, which had little effect on the result).

## 6. Conclusions

The presented studies showed the advantage of granulometric analysis over the other two methods of textural analysis (GLCM and Laplace filters) in the examined aspect. All tested variants increased the accuracy of classification in relation to the approach based only on spectral data. Also, almost all tested variants (with one exception) of the granulometric analysis showed greater efficacy than all the variants based on the other two methods of textural analysis.

The use of other tested methods of textural analysis, in the majority of analyzed cases, also increased the accuracy of classification. However, this is not a rule; in the case of both GLCM analysis and Laplace filtration, examples of deterioration in classification accuracy occurred. This suggests that, in some cases, the results of the textural analysis are irrelevant in distinguishing individual classes, which may partly be caused by the edge effect.

An additionally performed analysis of the random forest variable importance of components of the *spectral + GLCM* data set for random forest classification, showed the importance of Haralick's correlation. This type of analysis may be useful for the analysis of other GLCM statistics, but also for other methods of textural analysis. This may establish and test an optimal set of complementary textural data for the best possible increase in land use/cover classification accuracy.

**Funding:** This research received no external funding.

**Acknowledgments:** The author is grateful to Astri Polska for providing the Pleiades satellite imagery.

**Conflicts of Interest:** The author declares no conflict of interest.

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
