# Peer review of "The Comparison of Different Methods of Texture Analysis for Their Efficacy for Land Use Classification in Satellite Imagery"

_remotesensing, doi:10.3390/rs11101233_

Round 1

Reviewer 1 Report

There are comments about grammatical errors and word usage highlighted in yellow in attachment, and then some detailed comments are as follow. It is clear to me from the results section that the authors have done quite a bit of work, and I think that this area of research is of interest to the remote sensing community as very high spatial-resolution imagery becomes more widely available. However, the article lacks substantial detail in both the background information and the methods sections, and so I recommend accept it with major revisions.

Abstract

-          Ln 16: “different classes of forest and built-up areas” – either list out all classes, or if space is a concern, don’t list them and make the sentence more general.

-          Ln 20 “as tools providing textural data for classification” – rewrite this sentence to clarify.

-          Ln 22: “classification feature” – the term ‘feature’ is typically reserved for meanings other than its use in this sentence – suggest ‘as a classification element’ or ‘as a component of classification’

Introduction

-          Substantial English/ grammar edits required

-          L33: “As research shows…” This sentence needs supporting evidence - add references and more detail or rewrite

-          L41: delete “three of the above methods” and just list them. Delete everything after this sentence (Ln 443-49) and put this sentence in with the last paragraph

2.2 Brief presentation – section is misnumbered – should be 2.

-     2.1 – more references are needed in this section

-    2.2 – this section is FAR too short – more information (supported by references) should be given about Laplace filters

-     2.3 – this section contains a sufficient amount of background information about granulometric analysis, however figure 1 and 2 should be moved to the Results section.

3. Materials and Methods

Ln 106 – this should be moved to the introduction or not restated. A few general sentences should be included here to provide an overview of the methods. No details. Just general ‘workflow’ information.

Ln 112 – this information belongs in the background.

Ln 117 – ‘fragments’ is not the correct term – suggest ‘subset areas’

Table 2: suggest reformatting so that the change in rows can clearly be seen in the ‘images’ column.

Ln 125 – much more methodological information needed about the ‘processing of the image of the first principal component.’

In general -more detail in the methods section

4 Results

Suggest separating out results and discussion as 2 different sections.

Suggest sub-dividing the results section somehow… there are a LOT of results for the reader to wade through.

Ln 339 – What are units of the y axis?  How was ‘importance calculated’? This should be expanded/ clarified in the methods section.

Author Response

I want to thank for your thorough comments. I am convinced that they significantly improve the article. I hope that I have succeeded in taking them into account in a satisfactory way. 

Responses to individual comments can be found in the attached file.

Reviewer 2 Report

please see attached file.

Author Response

Thank you very much for the comments. I am sure, that they will improve the quality of the paper. I believe that I have included all of them in a satisfactory way.

Responses to individual comments can be found in the attached file.

Round 2

Reviewer 2 Report

My comments have been addressed.